# Analyzing the Occupied Space of Passengers with Reduced Mobility in Metro Station Platforms: An Experimental Approach Using a Tracking System

**Sebastian Seriani [1,\*], Pablo Guzman [2] and Taku Fujiyama [3]**

[1]   Escuela de Ingeniería de Construcción y Transporte, Pontificia Universidad Católica de Valparaíso, Valparaíso 2362804, Chile
[2]   Facultad de Ingeniería y Ciencias Aplicadas, Universidad de los Andes, Santiago de Chile 7620001, Chile
[3]   Faculty of Civil, Environmental and Geomatic Engineering, University College London, Gower St., London WC1E 6BT, UK
\*    Correspondence: sebastian.seriani@pucv.cl

**Abstract:** The objective of this study is to analyze the occupied space of passengers with reduced mobility when boarding or alighting a train through an experimental approach based on a virtual tracking tool system to obtain their exact position. The designed experiments considered a train and its adjacent platform, in which 21 volunteers were recruited, 3 with reduced mobility. The results indicated that passengers with reduced mobility required up to 80% more space, compared to a passenger without reduced mobility, when waiting to board the train. The passenger who occupied the largest space was the one with a pram, reaching 1.70 $m^2$/passenger, represented as a polygon. In addition, when passengers started to walk, the space used increased due to limb movement. In the alighting process, passengers with reduced mobility used almost twice the space required for the boarding process due to the relatively larger platform space occupied as each passenger alights and walks away, whereas when boarding, each passenger has less space to share with the other people waiting for the train to arrive or for the doors to open. These results could help practitioners improve the design of the platform or implement control measures, such as adding waiting areas for those passengers with reduced mobility. Further research is needed regarding other types of stations and density situations.

**Keywords:** micro-mobility; passenger; reduced mobility; space; tracking system; platform; metro station

## 1. Introduction

According to Rail Safety and Standards Board (RSSB) [1], four types of factors can affect the micro-mobility of pedestrians in public transport environments: the presence of other people (e.g., density on the platform), the physical design of the train carriage (e.g., the width of the doors and of the platform), the information provided to passengers (e.g., maps), and the environment (e.g., weather). In the case of metro stations, the presence of other people is considered the main factor, in which age, size, and culture affect the movement of pedestrians [2,3].

In order to study the micro-mobility while boarding and alighting a train in a metro station, it is necessary to define what is meant by pedestrian space. According to Fruin [2], a pedestrian in any standing area (e.g., a platform in a metro station) can be represented as an ellipse with an area 0.30 $m^2$, including a body depth of 50 cm and a shoulder breadth of 60 cm. However, when the pedestrian starts to walk, this area increases to 0.75 $m^2$ because there is additional space used for the movement of legs and arms [4].

This space is affected by the degree of crowding, especially in terminal stations in which a large number of passengers are boarding or alighting the train. For example, according to Tirachini et al. [5], the space in Santiago metro stations can be reduced to

0.16 m$^2$/passenger (i.e., 6 passengers/m$^2$) during peak hours, affecting passenger comfort and safety. Consequently, the same authors [5] suggest that the optimal density should be 3 passengers/m$^2$.

When there is a high level of crowding in metro stations, the sustainability of the city is affected. Aprigliano and Tirado [6] state that comfort and safety are essential to achieve the Sustainable Development Goals, especially with respect to Goal 11 (i.e., Sustainable Cities and Communities). Moreover, Unsworth et al. [7] reported that accessibility should be considered as a means to reach, and therefore use, the transport system, which could help improve quality of life, including for those passengers who need a mobility device to complete their journey. Therefore, the creation of good infrastructure, quality public spaces, and modes of transport that reach the most vulnerable and excluded people in society is essential to reduce the levels of inequality and poverty exhibited in Latin American cities [8]. One of the most vulnerable and excluded type of passengers is the passenger with reduced mobility. Recent studies concluded that in Chile, 16.7% of the population has some degree of disability (about 3 million people). Of the total, 38% are in the metropolitan region of Santiago which is the capital of the country [9]. Taking these figures into account, it is deemed necessary to carry out a study to consider the micro-mobility of passengers in public transport environments, such as metro stations.

This research work seeks to study the problem of micro-mobility in metro station platforms, understanding that the space used by passengers is a key element to understand their behavior, and therefore to improve the quality of transport services which could help to generate new standards for the design of these elements [10]. Even though in recent years, progress has been made in terms of accessibility on platforms and trains (e.g., installation of elevators in stations and placing more signs that demarcate reserved areas), the aim is to study the occupied space of different types of passengers with reduced mobility through full-scale experiments using a computational tool to extract their position.

The specific objectives are: (a) define the variables that affect the space occupied by different types of passengers; (b) propose a method to calculate the space used by them in the processes of boarding and alighting; (c) study audiovisual material obtained from experiments using a computational tool for the extraction of their positional coordinates within a dimensioned space; and (d) analyze the results for different types of passengers, considering the case of a terminal station in which a large number of passengers board or alight.

The article is structured in five sections. In Section 2, different studies related to the space used by passengers are described. In Section 3, the method is explained, followed by the results in Section 4. Finally, Section 5 presents the conclusions and discussion.

## 2. Existing Studies on Passenger Space in Metro Stations

Some manuals, such as the Highway Capacity Manual [10], use the concept of 'pedestrian' as any person that walks within the city (rural or urban areas). However, this study is focused on the concept of "passenger" (or pass), which is a person who uses the public transport infrastructure (e.g., metro stations).

This infrastructure should promote micro-mobility by generating accessibility to all types of passengers, especially to those who are typically excluded due to their limited capabilities (e.g., wheelchair users) [6,7]. However, achieving accessibility in metro stations is not a simple "check-list" exercise. For example, when the infrastructure presents a variation in the frequency and regularity of the services, a risk of cascading delays, or a "knock-on effect," are reached, in which trains cannot depart on time, affecting the accessibility of those passengers who need to wait on the platform for longer periods of time [11].

According to Gérin-Lajoie et al. [12], the space represented by each pedestrian can be obtained, considering those passengers surrounding them. This space can be defined as an elliptical area with a width of 0.96 m and a depth of 2.11 m, which is smaller when passing a static obstacle versus a moving obstacle. In addition, Gérin-Lajoie et al. [13] proved that

when going around a cylinder (or column) as an obstacle, the shape and edges (left and right) of this space can be asymmetric, where the longitudinal axis of the ellipse is related to speed, i.e., as a pedestrian's speed increases, the vertical axis becomes longer, while the horizontal axis is related to avoiding contact with other pedestrians or obstacles.

Such studies [12,13] are related to the concept of sensory zones, i.e., the distance that a person tries to maintain between their body and the rest of the environment, so that there is always enough time to perceive, evaluate, and react [14]. For example, for a normal walking speed, the sensory area can be estimated as a 1.06 m wide x 1.52 m deep elliptical area [14]. Fruin [2] has also calculated that at a normal walking speed of 1.37 m/s, the sensory zone reaches 1.48 m.

Other studies [15] reported that older adults require more space to move in high-density environments, but this increase may be moderated by social support and self-control. In this respect, Webb and Weber [16] argue that personal space is affected by vision, hearing, mobility, age, and gender (e.g., each pedestrian needs more space if mobility is limited). Those authors developed a theoretical model to understand the cognitive processes in a pedestrian's personal space based on the perception and interpretation of stimuli. In addition, Sakuma et al. [17] propose a simulation model based on psychological theory, according to which the personal space is defined by the inner critical circle, where any agent is immediately avoided, and the outer circle is discretely processed to avoid the appearance of pedestrians. The model includes the influence of individual memory on the decision to perform an action.

According to Daamen and Hoogendoorn [18], other characteristics of pedestrians, such as age, gender, and health status, should also be considered. In addition, the authors found that walking purpose, route knowledge, and luggage influence pedestrian behavior. Similarly, Willis et al. [19] indicated that in cities, men walk faster than women. The authors also found that age, travel conditions (e.g., bag, luggage), and time of day influence pedestrian behavior. Chataraj et al. [20] also include the effect of cultural differences, for example, the speed of Indian pedestrians is less affected by density than that of German pedestrians. The same authors [20] suggested the inclusion of local attitudes in relation to pedestrian behavior.

The proximity effect should also be mentioned as an important factor determining the relationship between micro-mobility and the used space. The first study was carried out by Hall [21], who divided the personal distance into four groups: (a) proximity (when the distance is less than 0.5 m and pedestrians have a special relationship); (b) personal zone (0.5~1.2 m and pedestrians know each other); (c) social consultation zone (1.2~4 m and pedestrians do not know each other, but allow communication); and (d) public distance (4 m to 10 m and pedestrians do not know each other and without communication or interaction). Regarding metro stations, Sommer [22] studied the social behavior of passengers and classified personal spaces using three groups: (a) intimate (<0.5 m); (b) personal (0.5–1.2 m); and (c) social (>3.0 m). Therefore, if the distance between the heads of two pedestrians is less than 1 m (consider 0.5 m plus twice the depth of Fruin's body ellipse [2]), then the pedestrians feel that their space has been encroached upon. However, this sense of intrusion is based on perception (e.g., comfort) rather than physical space (e.g., available space or density), which is difficult to calibrate on the platform in metro stations [5].

From another perspective, Schmidt and Keating [23] introduced the concept of personal control, which is less subjective than the personal space defined by Hall [21] and Sommer [22] and is divided into three types: behavioral, cognitive, and decisional. The first form is related to crowding situations where density interferes with pedestrian behavior or blocks pedestrian goals, or when pedestrians feel that they are losing control or spatial freedom (e.g., alighting passengers who cannot leave densely packed platforms). Regarding the cognitive aspects of travel, Schmidt and Keating [23] noted that individual control depends on how each pedestrian anticipates and interprets events or expected situations (e.g., stress). Advanced information can improve cognitive control (e.g., giving passengers a map of the busiest stations so they can plan their journey and reduce stress).

Finally, decision control refers to what an individual pedestrian expects when choosing an outcome.

Passengers in metro stations also try to avoid contact with other passengers unless such contact is unavoidable (for example, there is not enough space to board or alight) [2,3]. For example, Goffman [24] pointed out that to avoid collisions, pedestrians tend to form two streamlines, and the streamlines tend to be right-handed. For other authors [25–28], the way to avoid collisions depends on density and gender. In the case of subway stations, the interactions between boarding and alighting passengers can be seen as an extension of the social study of how people cooperate to avoid collisions. Recent laboratory experiments have used static obstacles to study collision avoidance techniques as a function of the visual field, where distances less than 1.5 m are considered difficult to avoid [29]. Previously, Fujiyama and Tyler [30] reported that pedestrians also scan each other to estimate their avoidance distance, which is approximately 5 m for a standing person. This behavior occurs on flat surfaces, but also on stairs.

In the case of corridors, concourses, and open areas, when pedestrians reach a high density, they auto-organize themselves and form lines of flow [31]. This phenomenon is produced only by the presence of other people and not because there are signals or markings. Some authors [2,3] identify that this phenomenon happens when the density is higher than 2 pass/m$^2$. Other authors suggest that this phenomenon is caused because pedestrians compete for their space, and they walk in groups (e.g., boarding or alighting) in which each pedestrian follows the pedestrian that is in front of them [32–34]. In this sense, Willis et al. [19] reported that a single pedestrian walks faster than a pedestrian with one or two companions. Moussaïd et al. [35] reported that groups are commonly composed of 2–4 members, and that there is an impact of the group on the crowd; for example, at low-density, members of the group tend to walk side-by-side, reaching a high speed; however, when density increases, speed decreases, and group forms "U" or "V" walking patterns, in which "V" patterns are the most efficient because of their aerodynamic shape.

Recently, some authors [36,37] proposed that passenger space can be represented as a polygon influenced by the surrounding passengers. Regarding the space used by a wheelchair passenger, the authors [37] found that wheelchair passengers on a platform waiting to board the train take up 61% more space than other passengers, considering the extra space used for assistance. Moreover, this space is affected by the density of passengers on the platform, i.e., as the platform density increases, the space used by passengers decreases [36,37].

Despite the important contributions and research reported in the literature review, new experiments are needed to study different types of passengers who use different mobility aids in the boarding or alighting process, which is the purpose of this study.

## 3. Experimental Method

The experimental method consisted of a full-size mock-up to represent the boarding and alighting of passengers in a terminal station of the metro in Santiago [37]. The set-up was a platform of 3.0 m long by 2.1 m wide (i.e., an area of 6.3 m$^2$), while the train had the same length as the platform and a width of 2.5 m.

In this mock-up, a total of 23 volunteers were recruited, of which 2 were located outside the platform to help with the opening/closing of the train doors, while the remaining 21 were part of the main experiment. With this, and considering the size of the platform, the average density was 3.33 pass/m$^2$, which is closer to the optimal condition in the Santiago metro, according to Tirachini et al. [5]. From the indicated density, the inverse can be applied, obtaining an average space available per person of 0.30 m$^2$/passenger.

In the experiments, different types of passengers with reduced mobility were studied:

- Firstly, among the participants, a 24-year-old young man was selected to move around with a pram.
- Secondly, an older adult, over 60 years of age with multiple complications, was selected. He presented a hearing problem and hemiparesis, which consists of an alteration of

movement and sensitivity that affects one side of the body, with the upper and lower extremities being the most affected. In addition, he had a deficit in coordination and balance, among other motor control problems of the body, while still able to walk.

- Thirdly, a young 24-year old man who used a wheelchair was chosen. He uses the wheelchair to move throughout his daily life, as he suffers from spastic diplegia, which consists of loss of strength in the lower limbs, accompanied by tension and rigidity in the musculature. The wheelchair model that he used was characterized by being light and self-propelled, which facilitates handling.

To evaluate the space used by each passenger with reduced mobility, reliable empirical data on their movement is needed for analysis and verification. Manual procedures for these types of exercises are very time-consuming and generally do not provide precision in space and time. For this reason, a pedestrian virtual tracking tool was used to automatically extract accurate trajectories from video recordings. With this tool, many people can be analyzed in an extensive series of different experiments. Due to the methods observed in previous studies [36,37], the Petrack software [38] was used. This software delivers the location of passengers in a given space in Cartesian coordinates.

The use of Petrack was considered under different conditions. First, passengers waiting to board the train were considered. In this case, passengers were standing on the platform according to a position (x, y, z) of their own choice. Secondly, a dynamic case was considered. In this situation, passengers were boarding the train, i.e., the space used by them considered the movement of their arms and legs when crossing the yellow safety line at the edge of the platform. Thirdly, another dynamic case was studied involving passengers alighting the train when crossing the yellow safety line at the platform edge. In all cases, the video camera was located at a height of 4 m on the ceiling of the laboratory to capture the image of the whole platform (see Figure 1).

Before obtaining the data, intrinsic and extrinsic calibrations were carried out to eliminate elements that may interfere with the tracking system. In the case of the intrinsic calibration, rotary movements (varying the inclination) of a chessboard-shaped grid were recorded using the same video camera as that used in the experiments. A total of 20 images of the chessboard were captured at different positions. This sequence of photos was entered into the Petrack software, which automatically arranged the frames. In this way, the curvature of the images was adjusted to correct the defect caused by the lenses of the video cameras. Having already parameterized the video curvature in the Petrack software, the external calibration was carried out. This seeks to map a coordinate system in the software that is equivalent to the environment in which the experiments were performed. With this, the software can generate a lattice and find an accurate position (x, y, z) of the participants.

To calculate the space occupied by a passenger with reduced mobility on the platform, specific criteria had to be defined that would allow for knowing the limits around the passenger. In this way, if the platform is unoccupied, the area in which a single person can move would be the total area of the platform (i.e., 6.3 m$^2$/passenger). However, if the density increases, the space used by each passenger is reduced. Therefore, it is necessary to define the boundary of the space occupied by each person in the presence of other passengers by using different criteria (the same criteria is used in the case of passengers without reduced mobility):

- The first criterion to consider passenger A as the boundary of the space used by passenger B was that passenger B should have direct visual contact with passenger A, without interference from another passenger. For this, the angle between a pair of passengers closer to passenger A must be greater than 5 degrees. For example, in Figure 2a, the angle between passenger N°2 and passenger N°3 is equal to 5 degrees, and it is possible for the passenger N°1 (with reduced mobility) to have direct full visual contact with passenger N°3, and therefore passenger N°3 is considered as a boundary of the area used by the passenger with reduced mobility.
- The second criterion establishes the distance between passengers. To be considered as a boundary of the space used by a passenger, a maximum distance between passengers

of 75 cm was assumed. For example, in Figure 2b, a pair of passengers (passengers N°2 and N°3) was considered as the boundary of passenger N°1, in which a triangular area was generated between them. In this case, an average body depth of 25 cm was assumed, following the methods of [2,3], and therefore, the distance between the shoulders of passengers is taken to be 50 cm.

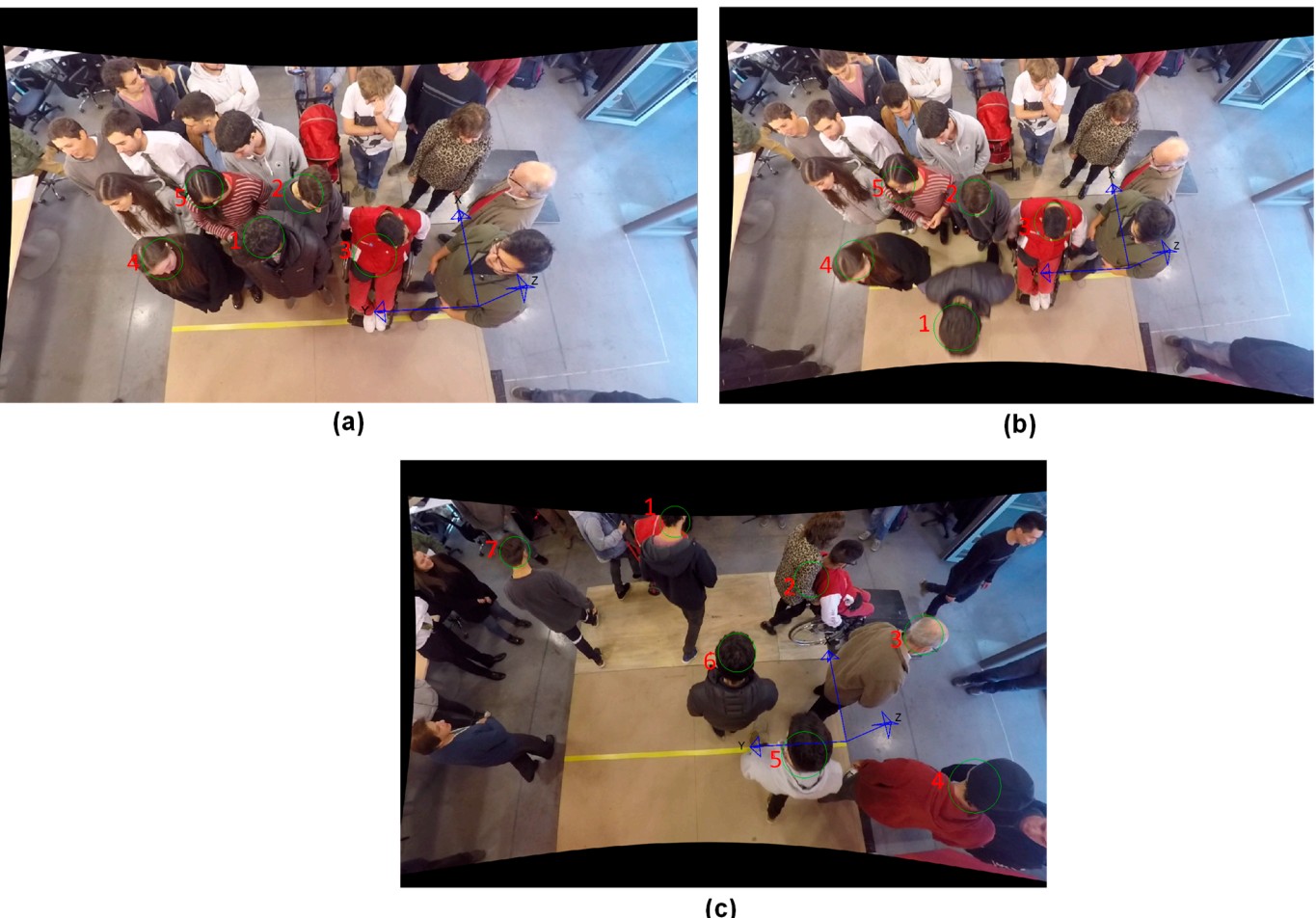

**(a)**

**(b)**

**(c)**

**Figure 1.** Platform used in the experiments at the laboratory in which passengers are detected (shown in green circle) and labeled (in red) from passenger N°1 to passenger N°*n*, where *n* is the last passenger around passenger N°1: (**a**) passengers waiting to board the train; (**b**) passengers boarding the train; (**c**) passengers alighting the train.

Having defined (using the criteria defined above) the points that describe the boundary of the space used by each passenger, the area is calculated for each case of study (and considering the three types of passengers with reduced mobility). With such a set of points, Petrack computes the Cartesian coordinates (horizontal and vertical), which indicate the position of each point on the floor plane.

In the case of a passenger with reduced mobility, Figure 3 shows an example of the location of a passenger using a pram according to the method described above. In this case, the space used by the passenger is represented as a polygon, in which the passenger with reduced mobility is located at the point (0, 0), using Petrack's reference origin set in the lower right corner of the platform (origin of blue axes in Figure 3). With these values, the angle and distance between users are calculated according to Equation (1).

$$d = \sqrt{(x_i - x_R)^2 + (y_i - y_R)^2}$$ (1)

where:

$x_i$ is the cartesian coordinate on the x-axis of participant $i$;

$y_i$ is the cartesian coordinate on the y-axis of participant $i$;

$x_R$ is the cartesian coordinate on the x-axis of the participant with reduced mobility;

$y_R$ is the cartesian coordinate on the y-axis of the participant with reduced mobility.

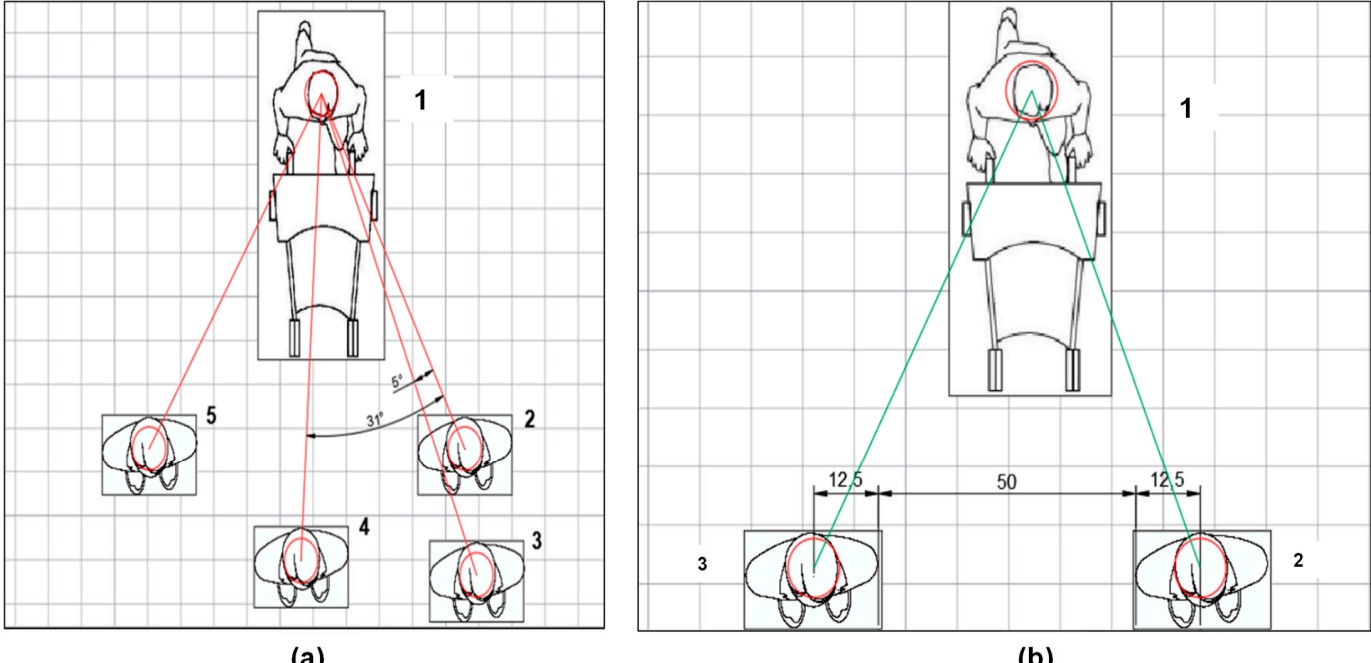

**Figure 2.** Criteria a passenger with the reduced mobility (labeled as passenger N°1) and those passengers around him/her (labeled as passengers N°2 to N°5): (**a**) according to the angle between them; (**b**) regarding the distance (in cm) between them.

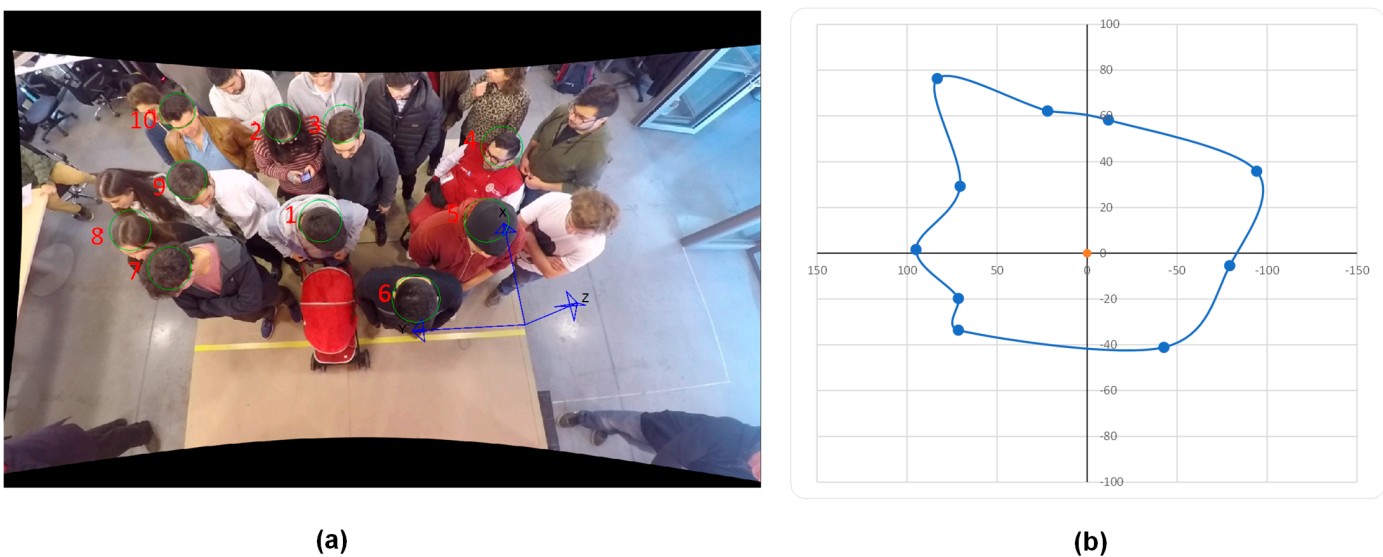

**Figure 3.** Example of the method used to identify a passenger with reduced mobility (passenger N°1) and those passengers around him/her (passengers N°2 to N°10): (**a**) passenger N°1 (pushing a pram) surrounded by 9 passengers (passengers N°2 to N°10); (**b**) dimensions in cm of the area used by the passenger N°1, who is located at (0,0).

As previously indicated, and given that the central objective of this work is to find the available space that a passenger occupies, triangular areas are formed within the generated polygons to completely fill the polygon's area. To obtain the area of each triangle, Equation

(2) is used. Finally, the sum of these areas provides the total space used by each passenger. Each of the triangular areas that make up the final polygon can be obtained from the determinant between the passenger and the surrounding passengers [37].

$$A_i = \frac{1}{2} \cdot \begin{bmatrix} x_R & y_R & 1 \\ x_i & y_i & 1 \\ x_{i+1} & y_{i+1} & 1 \end{bmatrix} \tag{2}$$

Once the relevant parameters (space used by a passenger and the distance to other passengers) for each person are obtained, a statistical analysis is carried out to observe the characteristics of those values, depending on the type of person being observed. In this way, they could be weighted if the means of the values show significant differences between them. Given the data obtained and considering that between 10 and 11 repetitions per experiment were carried out, the Mann–Whitney test was chosen, which is a non-parametric test, and the available parameters can be ordered from smallest to largest. In this case, 95% confidence was considered.

## 4. Results

The results are described in the following sections. Firstly, the calculation of the space used by passengers with and without reduced mobility is reported, considering the moment when they are waiting to board the train. Secondly, the occupied space is registered, considering passengers boarding or alighting the train. The analysis is based on a terminal station in which there is a high level of congestion on the platform.

### 4.1. Space Used by Passengers Waiting to Board the Train

The following results are presented in which the space occupied by a passenger with reduced mobility is compared to the case of a passenger without reduced mobility. The comparison is considered in the situation of passengers waiting to board the train on the platform.

Table 1 shows that the average space occupied by the wheelchair passenger varied in a range between 1.09 $m^2$/pass to 1.70 $m^2$/pass, with an average of 1.42 $m^2$/pass. The standard deviation is 0.26 $m^2$/passenger. It is interesting to note that the front distance is smaller than the rear distance, which could be due to passengers keeping a greater distance from the wheelchair, as the user needs more space to maneuver (e.g., make turns or stop before boarding the train). This is represented by distance *d*, that has an average value of around 70 cm.

**Table 1.** Average space occupied by a wheelchair passenger waiting to board the train.

| Runs | Occupied Space (m²/pass) | Rear Distance (cm) | Lateral Right Distance (cm) | Lateral Left Distance (cm) | Front Distance (cm) | Distance *d* to the Passenger with Reduced Mobility (cm) |
|---|---|---|---|---|---|---|
| 1 | 1.65 | 59.52 | 43.69 | 34.31 | 57.68 | 78.94 |
| 2 | 1.65 | 78.02 | 48.82 | 19.01 | 55.29 | 66.97 |
| 3 | 1.67 | 89.81 | 45.94 | 24.16 | 48.92 | 64.61 |
| 4 | 1.47 | 62.70 | 53.61 | 15.73 | 53.33 | 75.77 |
| 5 | 1.03 | 42.02 | 46.39 | 27.96 | 52.52 | 67.17 |
| 6 | 1.55 | 68.14 | 37.86 | 45.18 | 43.79 | 67.59 |
| 7 | 1.50 | 53.30 | 46.56 | 46.69 | 45.55 | 75.81 |
| 8 | 1.70 | 84.68 | 31.89 | 23.46 | 60.54 | 77.65 |
| 9 | 1.27 | 53.26 | 36.37 | 36.18 | 52.19 | 73.19 |

**Table 1.** *Cont.*

| Runs | Occupied Space (m²/pass) | Rear Distance (cm) | Lateral Right Distance (cm) | Lateral Left Distance (cm) | Front Distance (cm) | Distance $d$ to the Passenger with Reduced Mobility (cm) |
|---|---|---|---|---|---|---|
| 10 | 1.08 | 57.26 | 61.66 | 14.32 | 41.52 | 73.24 |
| 11 | 1.09 | 57.16 | 29.87 | 23.60 | 56.32 | 68.56 |
| Average | 1.42 | 64.17 | 43.88 | 28.23 | 51.60 | 71.77 |
| Standard Deviation | 0.26 | 14.61 | 9.41 | 11.06 | 6.03 | 4.96 |

The space occupied by the wheelchair user (1.42 m²/pass) represents 5 times the average value using Fruin's method [2] (0.30 m²/passenger, or level of service D). This average value using Fruin's method is obtained as the ratio between the platform area of 6.3 m² and 21 passengers, the total number of passengers in the experiments. This result is relevant, as it shows that a wheelchair user needs more space to move on the platform compared to the rest of the users.

In addition, it seems that the average value of density using Fruin's method [2] is not representative of the interaction of passengers on the platform. However, using the Petrack software [38], it is possible to locate where the wheelchair user is on the platform, and how many passengers surround the wheelchair user. Therefore, it would be possible to measure which part of the platform is more congested, and therefore, if an improvement in the design of the station layout is required or desirable (e.g., to increase the platform width).

In the case of the passenger with a pram, the average space is presented in Table 2. It is observed that the user with a pram occupied a greater space compared to the wheelchair user, with a significant difference of about 20%, according to the Mann–Whitney test, reaching an average of 1.70 m²/passenger. This is because the pram needs more space to maneuver, or requires assistance from a passenger.

**Table 2.** Average space occupied by a passenger using a pram waiting to board the train.

| Runs | Occupied Space (m²/pass) | Rear Distance (cm) | Lateral Right Distance (cm) | Lateral Left Distance (cm) | Front Distance (cm) | Distance $d$ to the Passenger with Reduced Mobility (cm) |
|---|---|---|---|---|---|---|
| 1 | 1.77 | 23.11 | 58.51 | 38.92 | 91.42 | 92.53 |
| 2 | 2.19 | 58.75 | 37.78 | 77.76 | 50.76 | 70.83 |
| 3 | 2.04 | 48.58 | 49.66 | 65.67 | 63.05 | 88.97 |
| 4 | 1.40 | 49.68 | 36.97 | 61.82 | 42.44 | 77.28 |
| 5 | 1.10 | 54.12 | 20.09 | 76.08 | 25.41 | 49.99 |
| 6 | 1.22 | 53.27 | 37.55 | 57.48 | 35.17 | 59.83 |
| 7 | 1.95 | 28.87 | 66.76 | 46.82 | 59.18 | 81.22 |
| 8 | 1.63 | 49.60 | 71.23 | 61.76 | 28.83 | 82.04 |
| 9 | 1.50 | 43.95 | 56.96 | 68.97 | 24.96 | 80.38 |
| 10 | 2.08 | 32.86 | 40.11 | 77.57 | 61.44 | 83.04 |
| 11 | 1.86 | 55.84 | 42.73 | 91.97 | 34.28 | 72.25 |
| Average | 1.70 | 45.33 | 47.12 | 65.89 | 46.99 | 76.21 |
| Standard Deviation | 0.37 | 11.84 | 15.10 | 15.05 | 20.51 | 12.47 |

Similarly, if the results are compared to the average values of density using Fruin's Level of Service, the difference between the average space used by a passenger with a pram (1.70 m²/passenger) is around 6 times the average space, using Fruin's method (0.3 m²/passenger). Therefore, the space occupied is determined by the type of passengers that wait to board the train. Consequently, greater spacing is generated, and therefore, the average distance between passengers and a passenger with reduced mobility increases.

The polygons generated by the space required by different passengers with reduced mobility are presented in Figure 4. Each polygon is represented in a different color, according to each run. The x-axis and y-axis represent the Cartesian coordinates of each passenger with reduced mobility who is at position (0,0). Figure 4a shows that the polygons of the user with a pram occupied a greater space compared to the wheelchair user (Figure 4b).

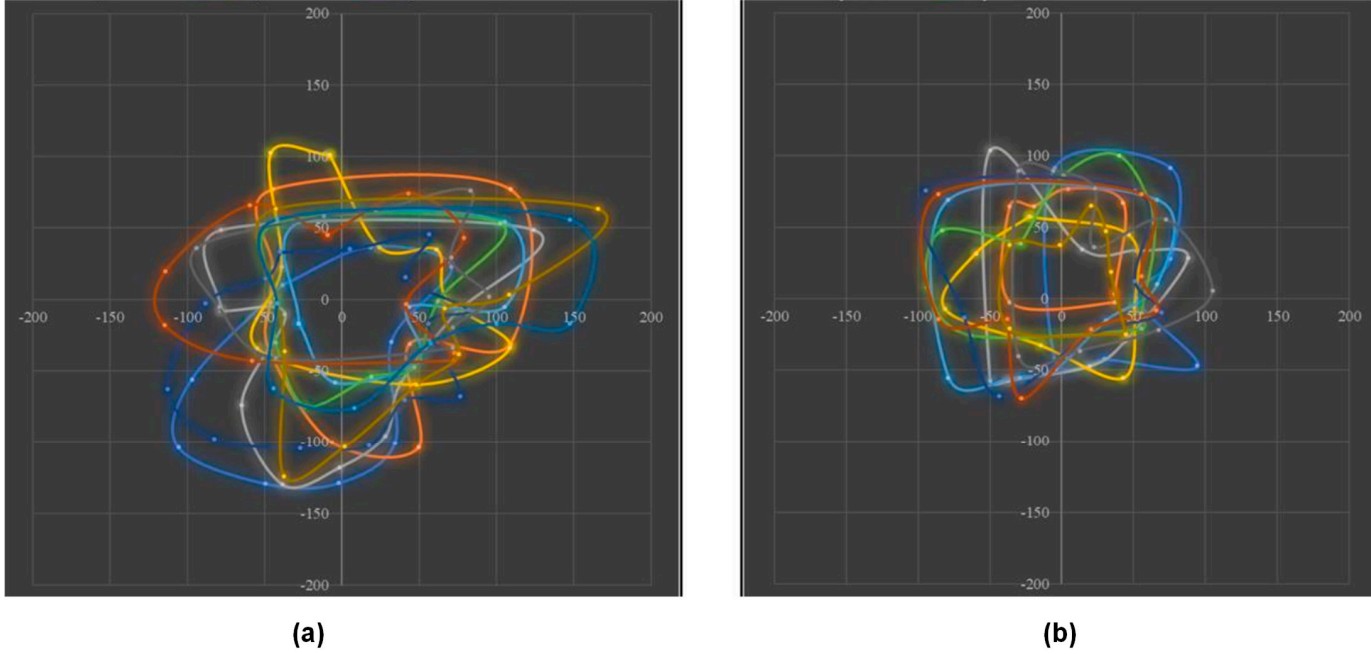

**(a)**      **(b)**

**Figure 4.** Bound graph represent the space used by a passenger with reduced mobility: (**a**) user with a pram; (**b**) wheelchair user. Coordinates are in cm.

With respect to the elderly person, Table 3 shows that this participant occupied a smaller space compared to the other two passengers with reduced mobility, reaching an average value of 1.1 m$^2$/passenger. For example, in the case of the wheelchair user, the difference is 23%, while in the case of the passenger with a pram, the difference is 35%. These variations in space use have statistical significance according to the Mann–Whitney test. Moreover, the passengers surrounding the elderly person exhibited a smaller distance compared to the wheelchair user (a reduction of 12%). Moreover, the space used by the elderly person (1.1 m$^2$/passenger) is about 3 times the average space using Fruin's Level of Service [2] (0.3 m$^2$/passenger).

**Table 3.** Average space occupied by an elderly passenger waiting to board the train.

| Runs | Occupied Space (m$^2$/pass) | Rear Distance (cm) | Lateral Right distance (cm) | Lateral Left Distance (cm) | Front Distance (cm) | Distance $d$ to the Passenger with Reduced Mobility (cm) |
|---|---|---|---|---|---|---|
| 1 | 1.04 | 52.95 | 4.79 | 65.40 | 43.88 | 72.38 |
| 2 | 0.55 | 55.37 | 30.50 | 30.14 | 2.69 | 65.25 |
| 3 | 1.15 | 50.07 | 45.69 | 43.65 | 48.08 | 60.89 |
| 4 | 0.96 | 44.58 | 44.60 | 46.67 | 25.25 | 61.76 |
| 5 | 1.53 | 49.37 | 47.74 | 54.59 | 43.86 | 53.44 |
| 6 | 1.28 | 55.64 | 67.08 | 43.28 | 19.08 | 79.96 |
| 7 | 1.51 | 54.77 | 68.63 | 47.31 | 29.46 | 60.93 |
| 8 | 1.43 | 54.08 | 59.35 | 49.16 | 25.72 | 53.89 |
| 9 | 1.14 | 58.88 | 22.07 | 50.50 | 32.72 | 74.06 |

**Table 3.** *Cont.*

| Runs | Occupied Space (m²/pass) | Rear Distance (cm) | Lateral Right distance (cm) | Lateral Left Distance (cm) | Front Distance (cm) | Distance $d$ to the Passenger with Reduced Mobility (cm) |
|------|------|------|------|------|------|------|
| 10 | 0.49 | 41.17 | 24.44 | 32.50 | 23.04 | 50.41 |
| Average | 1.11 | 51.69 | 41.49 | 46.32 | 29.38 | 63.30 |
| Standard Deviation | 0.36 | 5.45 | 20.85 | 10.15 | 13.62 | 9.69 |

In the case of the passengers without reduced mobility, Table 4 shows the average space occupied by them, which is in the range of 0.7 m²/pass to 1.09 m²/passenger, with an average of 0.94 m²/pass. Therefore, the space occupied by a passenger with a pram is 81% greater than that occupied by a passenger without reduced mobility. In the case of a wheelchair user, this figure is 51%. However, the case of the elderly person reached the smallest difference of 17%. Therefore, the results confirm that if the passenger with reduced mobility needs assistance or requires a mobility aid, then a bigger space is occupied on the platform.

**Table 4.** Average space occupied by passengers without reduced mobility waiting to board the train.

| Runs | Occupied Space (m²/pass) | Rear Distance (cm) | Lateral Right Distance (cm) | Lateral Left Distance (cm) | Front Distance (cm) | Distance $d$ to the Passenger without Reduced Mobility (cm) |
|------|------|------|------|------|------|------|
| 1 | 1.02 | 53.39 | 35.80 | 41.25 | 40.55 | 61.81 |
| 2 | 0.70 | 41.60 | 46.51 | 23.59 | 31.94 | 54.12 |
| 3 | 1.00 | 51.96 | 38.93 | 43.83 | 32.03 | 59.87 |
| 4 | 1.08 | 42.47 | 46.45 | 45.94 | 32.92 | 61.37 |
| 5 | 1.00 | 46.33 | 49.74 | 41.53 | 27.54 | 57.14 |
| 6 | 1.03 | 35.51 | 46.06 | 38.49 | 39.69 | 59.40 |
| 7 | 0.89 | 40.54 | 41.37 | 45.11 | 28.96 | 58.80 |
| 8 | 1.09 | 42.39 | 50.88 | 36.45 | 37.47 | 63.68 |
| 9 | 0.83 | 35.87 | 35.68 | 51.51 | 27.55 | 51.83 |
| 10 | 0.76 | 41.89 | 39.52 | 43.17 | 22.59 | 52.06 |
| 11 | 0.91 | 44.70 | 38.21 | 40.31 | 28.42 | 56.05 |
| Average | 0.94 | 43.33 | 42.65 | 41.02 | 31.79 | 57.83 |
| Standard Deviation | 0.13 | 5.64 | 5.47 | 7.04 | 5.59 | 3.96 |

### 4.2. Space Used by Passengers Boarding and Alighting

The following results are presented to compare the space used by a passenger without reduced mobility in the boarding and alighting processes. For this determination, five passengers were selected randomly from those who boarded or alighted the train. It is important to remember that the experiments represented the situation of a terminal metro station in which the train was empty when the boarding started, and the platform did not have passengers waiting to board when alighting started. In other words, passengers who alighted did not interact with passengers waiting to board the train. Similarly, passengers who boarded the train did not interact with passengers alighting. These interactions were out of the scope of this research; however, they should be considered in further studies.

Table 5 shows the average space occupied by a passenger without reduced mobility when crossing the yellow safety line at the platform edge. The average space occupied is 1.71 m²/passenger, which is comparable to the space occupied by a passenger using a pram who is waiting to board the train (see Table 2). In addition, the average space occupied by a passenger without reduced mobility boarding the train is 80% greater than the space

occupied by the same type of passenger waiting to board the train. In other words, when passengers are moving, they need more space for their legs and arms, which is in agreement with the study reported in [4].

**Table 5.** Average space occupied by passengers without reduced mobility boarding the train.

| Runs | Occupied Space (m²/pass) | Rear Distance (cm) | Lateral Right Distance (cm) | Lateral Left Distance (cm) | Front Distance (cm) | Distance $d$ to the Passenger without Reduced Mobility (cm) |
|---|---|---|---|---|---|---|
| 1 | 1.57 | 57.22 | 46.17 | 50.91 | 37.39 | 76.86 |
| 2 | 1.76 | 70.45 | 62.51 | 41.78 | 41.58 | 83.37 |
| 3 | 1.65 | 71.16 | 59.53 | 45.50 | 39.58 | 73.12 |
| 4 | 2.30 | 81.03 | 59.60 | 58.49 | 37.64 | 79.23 |
| 5 | 1.93 | 61.77 | 68.93 | 53.35 | 45.57 | 76.26 |
| 6 | 1.91 | 60.54 | 67.92 | 54.44 | 40.89 | 80.39 |
| 7 | 1.72 | 51.06 | 67.20 | 48.28 | 42.95 | 78.93 |
| 8 | 1.37 | 47.58 | 49.27 | 50.01 | 50.97 | 69.51 |
| 9 | 1.35 | 50.21 | 51.70 | 49.67 | 44.22 | 68.42 |
| 10 | 1.56 | 50.08 | 67.95 | 67.09 | 45.27 | 79.01 |
| 11 | 1.65 | 55.63 | 67.45 | 50.17 | 41.19 | 79.55 |
| Average | 1.71 | 59.70 | 60.75 | 51.79 | 42.48 | 76.79 |
| Standard Deviation | 0.61 | 19.44 | 16.36 | 16.43 | 9.18 | 13.41 |

In the case of the alighting process, Table 6 shows the average space occupied by passengers without reduced mobility when crossing the yellow safety line at the platform edge. The average space in the alighting process (3.74 m²/passenger) is almost twice the average space as in the boarding process (1.71 m²/passenger). This statistical difference may be explained by passengers being able to move along the platform to exit the experiment mock-up, reaching a lower density. However, in the boarding process, passengers were restricted by the train's structure (i.e., walls and train doors). This can also be noticed in the distance $d$ to the passenger without reduced mobility in which a value of 112.1 cm is reached (45% more than the distance as per the boarding process).

**Table 6.** Average space occupied by passengers without reduced mobility alighting the train.

| Runs | Occupied Space (m²/pass) | Rear Distance (cm) | Lateral Right Distance (cm) | Lateral Left Distance (cm) | Front Distance (cm) | Distance $d$ to the Passenger without Reduced Mobility (cm) |
|---|---|---|---|---|---|---|
| 1 | 2.63 | 103.77 | 83.53 | 72.39 | 36.05 | 102.74 |
| 2 | 3.67 | 98.95 | 78.17 | 87.29 | 55.84 | 97.22 |
| 3 | 3.64 | 102.55 | 70.56 | 97.20 | 58.82 | 125.53 |
| 4 | 4.06 | 106.38 | 76.20 | 82.59 | 60.92 | 118.45 |
| 5 | 3.65 | 125.99 | 85.74 | 84.51 | 42.23 | 104.22 |
| 6 | 4.60 | 125.06 | 100.34 | 69.43 | 60.88 | 109.80 |
| 7 | 4.17 | 108.96 | 101.21 | 60.18 | 62.79 | 122.18 |
| 8 | 3.76 | 104.91 | 105.82 | 68.31 | 55.48 | 105.78 |
| 9 | 3.49 | 117.09 | 91.05 | 65.24 | 53.17 | 112.20 |
| 10 | 2.91 | 100.40 | 89.71 | 54.75 | 48.57 | 115.21 |
| 11 | 4.52 | 114.16 | 125.03 | 64.19 | 47.16 | 119.81 |
| Average | 3.74 | 109.84 | 91.58 | 73.28 | 52.90 | 112.10 |
| Standard Deviation | 1.09 | 24.23 | 28.29 | 25.46 | 20.66 | 18.51 |

## 5. Conclusions and Discussion

This study was carried out in order to analyze the micro-mobility of passengers regarding the difference between the space occupied by passengers with and without reduced mobility when boarding or alighting the train through an experimental approach using a tracking system to obtain the position of each passenger in a controlled environment.

Based on the results, it can be concluded that a passenger with reduced mobility uses up to 80% more space than a person without reduced mobility when waiting to board the train. When passengers walk, this space increases due to the need to move their legs and arms, which is also reported in [2–4]. Boarding passengers use a smaller space compared with alighting passengers. This statistical difference is because passengers alighting could move through the platform to exit the mock-up; however, boarding passengers were restricted to the layout of the train by physical design features such as train doors and a hall entrance layout, which could affect the interaction of passengers [36].

The space occupied by passengers was represented by polygons rather than ellipses, making it more complex to parameterize. As reported in [12,13], under normal conditions, space can be represented by an ellipse; however, in the case of passengers with reduced mobility, this is represented as a polygon without a standardized shape. For example, in the case of the elderly passenger, in each of the repetitions of the experiments, it was possible to perceive that he was afraid of boarding the train through the center of the platform, as most passengers did. In fact, the elderly passenger at the moment prior to boarding the train tended to stay behind or at the side of the mass of users who boarded the train.

Considering the differences in the space used by each passenger, the type of passenger who required the most space was the one who used a pram, reaching 1.70 $m^2$/pass. This leads us to think that for similar examples of personal occupation (e.g., people with a suitcase, a guide dog, or a cane, or those accompanied by a child, etc.), there could be a pattern that determines that the area needed to move varies proportionally to the difference in the occupied area.

This type of study could lead to the investigation of the congestion at the platform and generate extra preferential waiting areas based on the study of passengers with reduced mobility on metro station platforms. In addition, it is urged to generate greater visibility of the waiting areas, to redesign the space, and to generate more markings or signs at key points to warn the population or guide traffic to relevant areas. It is also proposed to generate campaigns to educate users using the information screens in metro stations. In addition, greater control at peak hours, making use of waiting areas, is needed. Finally, access to these areas through separators or special entrance areas prior to moving to the platform needs to be improved in stations.

The analysis presented here is also relevant because, in extraordinary situations, such as the one experienced in connection with the COVID-19 pandemic, the measurement of occupied space turns out to be important for the management of this type of crisis. This also reveals the need to develop a culture in national public transport that improves the way users behave, as this would make it easier consider passengers with reduced mobility and to respect intimate spaces, thus reducing the stress caused by the feeling of overcrowding, as reported in [15,16,21,22].

Regarding future lines of research, it would be interesting to carry out more experiments to study boarding and alighting simultaneously (e.g., passengers alighting who interact with those who are boarding the train). Furthermore, this same study should be replicated based on the current operating conditions of metro stations, that is, of a higher occupational density (e.g., 6 passengers/$m^2$).

**Author Contributions:** Conceptualization, S.S. and T.F.; methodology, S.S. and T.F.; software, S.S. and P.G.; validation, S.S.; formal analysis, P.G.; investigation, S.S., P.G. and T.F.; resources, S.S.; data curation, P.G.; writing—original draft preparation, S.S.; writing—review and editing, P.G. and T.F.; visualization, P.G.; supervision, S.S. and T.F.; project administration, S.S.; funding acquisition, S.S. All authors have read and agreed to the published version of the manuscript.

**Funding:** This research was funded by FONDECYT 11200012 and FONDEF id22i10018, research projects from ANID, Chile.

**Institutional Review Board Statement:** The study was conducted according to the guidelines of the Declaration of Helsinki and approved by the Institutional Review Board (or Ethics Committee) of Universidad de Los Andes (protocol code CEC202089, approved on 23 October 2020) and Pontificia Universidad Católica de Valparaíso (protocol code BIOEPUCV-H 548-2022, approved on 3 October 2022).

**Informed Consent Statement:** Informed consent was obtained from all subjects involved in the study.

**Data Availability Statement:** Not applicable.

**Acknowledgments:** The author would like to thank the volunteers who simulated the boarding and alighting process in the laboratory facility. In addition, the authors are thankful to Maik Boltes, who helped to calibrate the Petrack software used in the study.

**Conflicts of Interest:** The authors declare no conflict of interest.

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
