# Peer review of "Analyzing the Occupied Space of Passengers with Reduced Mobility in Metro Station Platforms: An Experimental Approach Using a Tracking System"

_applsci, doi:10.3390/app13031895_

Round 1

Reviewer 1 Report

* Organization of whole paper from abstract to conclusion need to be improved.
* There are several typo mistakes and grammatical issues throughout the manuscript.
* Methods used in this manuscript are not clearly implemented.
* Results and discussion portion needs more comprehensive explanations based on study point.
* Tables should be revised in an articulated way.

Consider these papers in your manuscript:

Maturity evaluation of supply chain procedures by combining SCOR and PST models

Using fuzzy DEMATEL and fuzzy Similarity to develop a human capital evaluation model

Mathematical modeling of Green closed loop supply chain network with consideration of supply risk: Case Study

Author Response

Dear Reviewer,

Thank you very much for your comments, we tried to address them as best as possible.

Best wishes,

Sebastian

Reviewer 2 Report

The document is very good readable with a high understable description of the research activity. 

The weak point of the Document is the definition of goals to achieve from the description of the work done and its results. As an example, it seems that all the work done quantifies the area of the platform used by an user (in particular a PRM) without any other conclusion (for instance a comparison to other studies or real test campaigns)

The systematic approach followed by the authors, particularly for PRM, does not promote any other consequence than the evidence of the necessary area for an user and a quantification of this. Then, a question is in mind: are this result supported by real tests or complementary studies? 

In other words, it cannot obtain any conclusion to understand what are the consequences of the analysis on the metro platforms (bottleneck analysis, operations, functions, evacuation analysis, among others).

As conclusion from the above appreciations, a whole and deep review is recommended.

Author Response

(The authors gave the same response as above.)

Reviewer 3 Report

The authors propose a micro-mobility analysis method in metro station and analyze the corresponding effects. Overall, this paper is well-organized, there are some minor concerns which need to be improved in the revised version:

1. Abstracts: "a tracking tool system" what kind of technology is applied in the system? for example, the author needs to specify "a visual tracking system" or others.

2. Part 2: The author needs to discuss the disadvantages of existing works and the contributions of your work in detail in the last paragraph.

3. In the experiment part 3, what are the effects of different range of ages in the final results? only the people over 60 and smaller than 24 are discussed.

4. Results Part: An overall designed process for result analysis need to be described at the beginning of this part.

5. The full form is required when abbreviation is used in the paper for the first time, eg,. RSSB, 

6. There are missing comparison or discussion with the similar algorithm in the result part.

Author Response

(The authors gave the same response as above.)

Round 2

Reviewer 1 Report

accept